# KOROL: Learning Visualizable Object Feature with Koopman Operator Rollout for Manipulation

**Hongyi Chen**[1], **Abulikemu Abuduweili**[1]*, **Aviral Agrawal**[1]*, **Yunhai Han**[2]*,
**Harish Ravichandar**[2], **Changliu Liu**[1], **Jeffrey Ichnowski**[1]
[1] Carnegie Mellon University, [2] Georgia Institute of Technology

**Abstract:** Learning dexterous manipulation skills presents significant challenges due to complex nonlinear dynamics that underlie the interactions between objects and multi-fingered hands. Koopman operators have emerged as a robust method for modeling such nonlinear dynamics within a linear framework. However, current methods rely on runtime access to ground-truth (GT) object states, making them unsuitable for vision-based practical applications. Unlike image-to-action policies that implicitly learn visual features for control, we use a dynamics model, specifically the Koopman operator, to learn visually interpretable object features critical for robotic manipulation within a scene. We construct a Koopman operator using object features predicted by a feature extractor and utilize it to auto-regressively advance system states. We train the feature extractor to embed scene information into object features, thereby enabling the accurate propagation of robot trajectories. We evaluate our approach on simulated and real-world robot tasks, with results showing that it outperformed the model-based imitation learning NDP by $1.08\times$ and the image-to-action Diffusion Policy by $1.16\times$. The results suggest that our method maintains task success rates with learned features and extends applicability to real-world manipulation without GT object states. Project video and code are available at: https://github.com/hychen-naza/KOROL.

**Keywords:** Manipulation, Koopman Operator, Visual Representation Learning

## 1 Introduction

Humans possess an extraordinary ability to manipulate objects, discerning position, shape, and other properties with just a glance. How can robots be endowed with similar perceptual and dexterous manipulation capabilities? Traditional control and optimization approaches typically require detailed models of the system dynamics [1, 2]. However, these models can be difficult to derive and often lack the flexibility and generalizability needed to adapt to task or environment changes. End-to-end data-driven methods overcome these challenges by learning actions directly from observations [3, 4, 5]. While these methods can make minimal assumptions, they often require a large number of demonstrations to master basic skills due to the high dimensionality of the inputs.

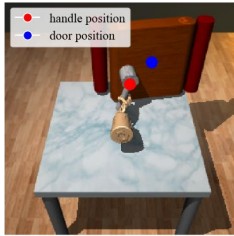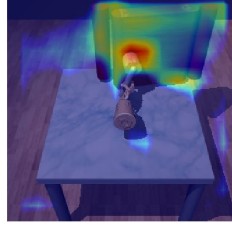

**Vanilla Koopman Operator** **KOROL (Ours)**

Figure 1: **left:** Vanilla Koopman operators rely on ground-truth state which may be difficult to obtain in real-world settings. **right:** In contrast, we propose KOROL, which learns a dynamics model and task-relevant object features without labels of object states. The visualization shows the localization of learned feature around the door handle.

To combine sample efficiency of traditional model-based approaches with high generalizability of deep learning methods, one branch of recent work has focused on learning dynamics models to plan

8th Conference on Robot Learning (CoRL 2024), Munich, Germany.

---

* Denotes equal contribution

trajectories. These methods embed learning into various models, such as Koopman operator [6, 7], Dynamic Movement Primitives (DMP) [8], Neural Geometric Fabrics [9, 10], and more [11], showcasing good performance in simulations. However, they often falter in real-world applications due to their reliance on hard-to-obtain ground-truth (GT) state information like object poses and contact points. Moreover, the problems of using computer vision to estimate these object states are the uncertainty in determining the number of objects to consider and which specific states to estimate. Additionally, the learned dynamics models do not transfer across different tasks without a universal state space design.

We propose an approach to remove the dependency on GT states in model-based manipulation learning (Figure 1). Central to this approach, **K**oopman **O**perator **R**ollout for **O**bject Feature **L**earning (KOROL), is learning visual features that predict robot states during dynamics model rollouts. Unlike learning methods that learn implicit visual features for image-to-action policies, KOROL explicitly trains on visual object features, encoding essential scene information to enhance predictions of robot states during autoregressive model rollouts. Central to this rollout process is the Koopman operator, which utilizes current object features to advance robot states. This establishes a synergistic relationship between the learned object features and the Koopman operator. KOROL uses trained features to refine the Koopman operator, which in turn improves the feature learning process through a more accurate dynamics model.

In experiments, we demonstrate how KOROL outperforms prior methods in ADROIT Hand [12] and generates interpretable visualizations by learning object features from images. We also show how KOROL enables the application of Koopman operator in vision-based real-world manipulation tasks. Finally, we demonstrate how learning object feature instead of designing object states for dynamics modeling enables KOROL to construct universal Koopman dynamics across multiple manipulation tasks. This paper makes the following contributions:

- We introduce KOROL, an imitation learning method, that uses the Koopman operator to learn object features and show that the Koopman operator with learned object features can outperform the one with GT object states.
- We extend the application of the Koopman operator to vision-based manipulation tasks in real-world by learning object features from images and demonstrate its effectiveness through comparisons with prior methods.
- We demonstrate that KOROL learns dimensionally-aligned object features across tasks, enabling the development of a multi-tasking Koopman operator.

## 2   Related Work

**Imitation Learning and Visual Representations for Manipulation.**   Imitation learning serves as a primary method for teaching robots to manipulate objects by mapping observations or world states directly to actions. Common approaches include Behavioral Cloning (BC) [13], Implicit Behavioral Cloning (IBC) [14], Long Short-Term Memory (LSTM) networks [3, 15], Transformers [4, 5], and Diffusion Models [16]. A significant challenge in imitation learning is representing the visual information of a scene. Strategies include using pre-trained 2D [17] or 3D backbones [5] to output visual embeddings. Other works propose end-to-end learning approaches that simultaneously train the visual encoder and the learning policy [4, 16]. Other techniques focus on learning visual representations through correspondence models [15], self-supervised novel view reconstruction using Neural Radiance Fields (NeRF) [18, 19], and Gaussian Splatting [20].

**Model-Based Learning and Planning.**   In robotics, traditional model-based approaches rely on expert knowledge of physics to design system models [21, 22, 2]. Since traditional methods can miss complex nonlinearities, and end-to-end learning approaches can be data-intensive, a middle ground of data-driven model learning shows promise as a data-efficient way to derive complex models [6, 23]. Model learning includes a variety of dynamics models such as Koopman operators [24, 25], Deep Neural Koopman operators [7], Dynamic Movement Primitives [8], Neural Geometric Fabrics [9, 10], and others [11, 26]. Additionally, some studies focus on learning environmental responses to actions to plan a future trajectory [27, 28, 29], integrate planning in a generative modeling process [30, 16], and seamlessly blend the learning of models with planning [31, 32].

**Koopman Operator Theory.** In the early 1930s, Koopman and Von Neumann introduced the Koopman operator theory to transform complex, nonlinear dynamics systems into linear ones in an infinite-dimensional vector space, using observables as lifted states [33, 34]. This transform allows the application of linear system tools for effective prediction, estimation, and control with hand-designed observables [6, 35, 36, 37]. Recent methods using neural networks to learn observables have proven more expressive and effective, particularly in chaotic time-series prediction [7, 38, 39]. Furthermore, the integration of neural network-derived Koopman observables with Model Predictive Control has shown promise in enhancing control tasks [40, 41]. As a significant benchmark, Han et al. [6] demonstrate the effectiveness of Koopman operators in manipulation tasks using GT object states. Building on this foundation, we extend the application of the Koopman operator to vision-based manipulation tasks in real-world settings by learning object features directly from images

## 3 Background: Koopman Operator Theory

In this section, we provide a brief background on the Koopman Operator Theory. Consider the evolution of nonlinear dynamics system $x(t + 1) = F(x(t))$. Given the original state space $\mathcal{X}$, the *Koopman Operator $\mathcal{K}$* introduces a lifted space of *observables $\mathcal{O}$* using *lifting function $g : \mathcal{X} \to \mathcal{O}$*, to transform the nonlinear dynamics system into a linear system in infinite-dimensional observables space as $g(x(t + 1)) = \mathcal{K}g(x(t))$.

In practice, we approximate the Koopman operator by restricting observables to be a finite-dimensional vector space. Let $\phi(x(t)) \in \mathbb{R}^p$ represent a finite dimensional approximation of observables $g(x(t))$, and a matrix $\mathbf{K} \in \mathbb{R}^{p \times p}$ approximate the Koopman operator $\mathcal{K}$. Thus, we rewrite the relationship as

$$\phi(x(t + 1)) = \mathbf{K}\phi(x(t)). \tag{1}$$

Given a dataset $D$, in which each trajectory $\tau = [x(1), x(2), \cdots, x(T)]$ containing $T$ time steps, we can learn $\mathbf{K}$ by minimizing the state prediction error [38]

$$\mathbf{J}(\mathbf{K}) = \sum_{x \in D} \sum_{t=0}^{t=T-1} \|\phi(x(t + 1)) - \mathbf{K}\phi(x(t))\|^2. \tag{2}$$

In manipulation tasks, we define the state $x(t) = [x_r(t)^\top, x_o(t)^\top]^\top$ to include the robot state $x_r(t)$ and object state $x_o(t)$, as we care about how objects move as a result of robot's motion. Moreover, since our goal is to minimize the imitation error of the robot state $x_r(t)$, we design observables $\phi(x(t))$ that include lifted robot and object states as

$$\phi(x(t)) = [x_r(t)^\top, \psi_r(x_r(t)), x_o(t)^\top, \psi_o(x_o(t))]^\top \ \forall t, \tag{3}$$

where $\psi_r : \mathbb{R}^n \to \mathbb{R}^{n'}$ and $\psi_o : \mathbb{R}^m \to \mathbb{R}^{m'}$ are vector-valued lifting functions that transform the robot and object state respectively. We can thus retrieve the desired robot state by selecting the corresponding elements in $\phi(x(t))$. Let $\phi^{-1}$ denote the *unlifting* function to reconstruct the robot state from observables, $x_r(t) = \phi^{-1} \circ \phi(x(t))$ (we can also reconstruct the object state $x_o(t)$ in the same way). Considering the lifting function Equation 3, the unlifting function can be represented as

$$x_r(t) = \phi^{-1} \circ \phi(x(t)) = [I_{n \times n}, 0_{n \times (n'+m+m')}] \cdot \phi(x(t)), \tag{4}$$

where $I_{n \times n}$ and $0_{n \times n}$ denote an identity matrix and zero matrix respectively. To streamline notation throughout this paper, we define $\hat{x}_r(t + 1) = \mathbf{K}'(x_r(t), x_o(t))$, where $\mathbf{K}' := \phi^{-1} \circ \mathbf{K} \circ \phi$.

## 4 Method

We propose KOROL, which formulates dynamics learning and object feature learning as an imitation (supervised) learning problem on robot states. Given a dataset $D$, in which each trajectory $\tau = [x_r(1), y(1), x_r(2), y(2), \cdots, x_r(T), y(T)]$ containing robot states $x_r(t)$ and image observation $y(t)$ of the object, instead of object state $x_o(t)$, our goal is to learn a visual object feature extractor $f_\theta$ and a Koopman operator $\mathbf{K}$ which can predict object features from images that minimize the robot states imitation errors. In this formulation, (2) becomes

$$\underset{\theta, \mathbf{K}}{\arg\min} \sum_{x \in D} \sum_{t=0}^{T-1} \|x_r(t + 1) - \mathbf{K}'(x_r(t), f_\theta(y(t)))\|^2. \tag{5}$$

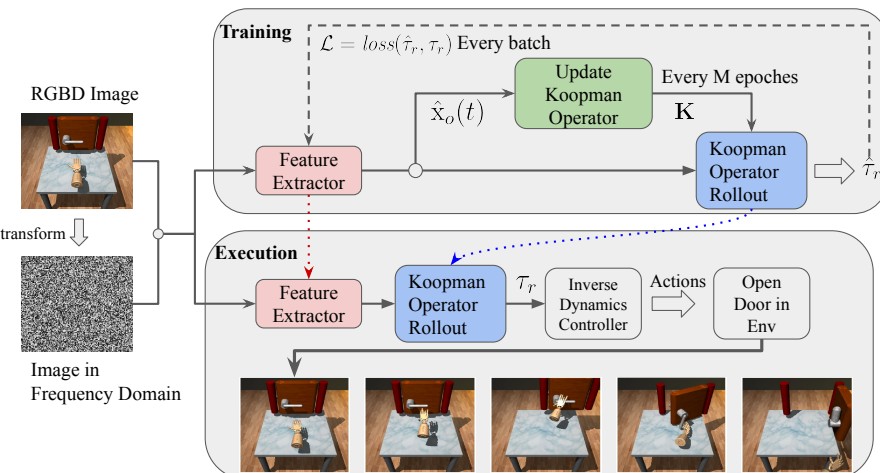

Figure 2: **Training and Execution Pipeline.** During training, KOROL updates the feature extractor $f_\theta$ based on the loss between the predicted robot trajectory $\hat{\tau}_r = [\hat{x}_r(1), \hat{x}_r(2), \cdots, \hat{x}_r(T)]$ obtained through Koopman operator rollouts and the ground-truth robot trajectory $\tau_r = [x_r(1), x_r(2), \cdots, x_r(T)]$. KOROL updates the Koopman operator with the new object features $\hat{x}_o(t)$ every $M$ epochs to enhance the training of $f_\theta$. During execution, KOROL feeds the generated trajectory to the inverse dynamics controller to produce the actions.

**Learning object feature.** While traditional Koopman operator construction requires GT object state information $x_o(t)$ (section 3), instead, we adopt a neural network $f_\theta$ for object feature encoding and extraction from RGBD images. We initialize $f_\theta$ and predict object features $\hat{x}_o(t)$ for all images in $D$ to construct the initial Koopman operator $\mathbf{K}$, as proposed by Han et al. [6]. During training, we randomly sample the beginning time step $t_0$ in trajectory $\tau$, and select $\hat{x}_o(t_0) = f_\theta(y(t_0))$. Then we can advance the observables forward using $\mathbf{K}\phi(\cdot, \cdot)$. We train $f_\theta$ by minimizing the loss function

$$\mathcal{L} = \mathbb{E}_{\tau, t_0} \left[ \sum_{i=0}^{N-1} \| x_r(t_0 + i + 1) - \mathbf{K}'(\hat{x}_r(t_0 + i), \hat{x}_o(t_0 + i))) \|^2 \right], \tag{6}$$

where $N$ is the prediction horizon and $\hat{x}_r(0) = x_r(0)$ indicates that the system provides the initial GT robot state. See Figure 2 for visualization. Integrating spatial domain RGBD images with their frequency domain counterparts has been shown to enhance image classification performance by accentuating discriminative features [42, 43]. Therefore, we apply the Discrete Cosine Transform [44] to convert RGBD images into the frequency domain. We then concatenate the spatial and frequency domain images as input, enabling $f_\theta$ to detect changes in successive, highly-correlated images more effectively than using spatial images alone. Subsequently, KOROL generates the reference trajectory ($\{\hat{x}_r(t)\}_{t=1}^N$) by rolling out the dynamics $\mathbf{K}$. We feed these trajectories into the pre-trained inverse dynamic controller [6], which computes the required action $a(t)$ using $\hat{x}_r(t)$ and $\hat{x}_r(t+1)$.

**Updating of Koopman Operator K.** Subsequent updates to the Koopman operator $\mathbf{K}$ are necessitated by changes in the predicted object features $\hat{x}_o(t)$, because $\mathbf{K}$ is optimized for these specific robot states and object features. See pseudocode in Alg. 1 for details. During the training of $f_\theta$ from line 7 to line 15, the dynamics $\mathbf{K}$ initially computed at line 3 may no longer be optimal for the new object features, prompting a need for recalculation. However, recalculating the object features across the entire training dataset and updating $\mathbf{K}$ for every $f_\theta$ modification is computationally intensive. Therefore, in KOROL, we defer the updates and recalculate $\mathbf{K}$ every $M$ epoches to balance accuracy with computational efficiency, as detailed from line 16 to line 20.

**Multi-tasking Koopman Operator.** While a robot's state space remains consistent when using the same robot platform, object state spaces typically vary across different tasks. For example, a prior Koopman manipulation study [6] includes a 15-DoF *tool use* task and a 7-DoF *door opening* task. Due to the differences in object state space definition and dimensions, Koopman operators trained for different tasks can not be shared, limiting their scalability. In KOROL, we propose training object features $\hat{x}_o(t)$ to serve as a universal interface for representing length-varied object states

---

**Algorithm 1** Object Feature Learning and Koopman Operator Updating

---
1: **Require** training dataset $D$ with robot states $x_r$ and images $y$, feature extractor $f_\theta$, function for calculating Koopman operator $func(\cdot)$
2: $\hat{x}_o \leftarrow f_\theta(y)$ for $y$ in $D$ // Predict object features across dataset
3: $\mathbf{K} \leftarrow func(x_r, \hat{x}_o)$ // Calculate the initial dynamics $\mathbf{K}$
4: **for** epoch$= 1, \ldots, N_1$ **do**
5: $\quad$ $\tau, t_0 \sim D$ // Sample the trajectory and beginning time steps
6: $\quad$ $x_r(t_0), y(t_0) \leftarrow \tau(t_0)$
7: $\quad$ $loss \leftarrow 0$
8: $\quad$ **for** $i = 0, \ldots, N$ **do**
9: $\quad\quad$ **if** $i = 0$ **then**
10: $\quad\quad\quad$ $\hat{x}_r(t_0) \leftarrow x_r(t_0); \hat{x}_o(t_0) \leftarrow f_\theta(y(t_0))$ // Predict object feature with feature extractor
11: $\quad\quad$ **end if**
12: $\quad\quad$ $\hat{x}_r(t_0 + i + 1) \leftarrow \mathbf{K}'(\hat{x}_r(t_0 + i), \hat{x}_o(t_0 + i))$ // Predict the next states with $\mathbf{K}$
13: $\quad\quad$ $loss \leftarrow loss + \|x_r(t_0 + i + 1) - \hat{x}_r(t_0 + i + 1)\|^2$ // Calculate and sum the loss using Equation 6
14: $\quad$ **end for**
15: $\quad$ Update the feature extractor $f_\theta$ to minimize $loss$
16: $\quad$ **if** epoch % $M = 0$ **then**
17: $\quad\quad$ $\hat{x}_o \leftarrow f_\theta(y)$ for $y$ in $D$
18: $\quad\quad$ $\mathbf{K} \leftarrow func(x_r, \hat{x}_o)$ // Update the Koopman operator
19: $\quad$ **end if**
20: **end for**

---

across tasks, enabling the generalization of $\mathbf{K}$ to multiple tasks. Moreover, these object features act as latent conditional vectors that differentiate among tasks. Thus, as long as the feature extractor can identify useful object features, it is possible to use datasets from various tasks to train a single multi-task Koopman operator $\mathbf{K}_{\text{multi}}$.

## 5 Experiments

In this section, we evaluate the performance of KOROL along with existing unstructured learning and model-based learning approaches in simulation and real-world tasks.

### 5.1 ADROIT Hand Simulation Experiment

**Setup and Baselines.** We conducted our simulation experiments on the ADROIT Hand [12]—a 30-DoF simulated system (24-DoF articulated hand + 6-DoF floating wrist base). There are 4 simulation tasks: *Door opening*, *Tool use*, Object *Relocation*, and In-hand *Reorientation*. We compared KOROL to the baselines: (1) *Behavior Cloning (BC)*: Unstructured fully-connected neural network policy; (2) *Neural Dynamic policy (NDP)*: Neural network policy with embedded structure of dynamics systems [8]; (3) *Diffusion Policy*: Learning policy using probabilistic generative model [16]. To allow equal comparison, all models use ResNet18 [45] as feature extractor. Appendix provides details about task state space design and baselines implementation.

| Model | Door opening | | Tool use | | Relocation | | Reorientation | |
|---|---|---|---|---|---|---|---|---|
| | 10 | 200 | 10 | 200 | 10 | 200 | 10 | 200 |
| BC w GT | 0% | 96.1% | 0% | 49.5% | 0% | 48.1% | 19.4% | 67.8% |
| NDP w GT | 5.2% | 99.9% | 30.2% | 96.9% | 1.9% | 99.8% | 21.6% | 64.6% |
| Diffusion Policy w GT | 97.5% | 100% | 99.4% | 100% | 59.6% | 99.2% | 83.8% | 93.3% |
| Koopman Operator w GT | 99.6% | 100% | 100% | 100% | 77.0% | 95.6% | 7.6% | 83.6% |
| BC | 0% | 0% | 0% | 0% | 0% | 0% | 0% | 0% |
| NDP | 0% | 99.3% | 0% | 96.2% | 0% | 92.7% | 25.3% | 67.7% |
| Diffusion Policy | 93.2% | 99.9% | **97.8%** | 99.7% | 86.4% | 100% | 31.5% | 33.0% |
| KOROL | **98.6%** | 99.9% | 94.3% | **100%** | **99.8%** | 100% | **55.6%** | **86.4%** |

Table 1: **Quantitative Performance in ADROIT Hand.** The averaged task success rates across 5 random seeds for all models, trained with either 10 and 200 demonstrations per task. We evaluated each model on 200 unseen cases per task. The upper half of the table displays results for models using GT object states, while the lower half displays results from models employing features extractor ResNet18.

**Numerical Results of KOROL and Baselines.** From the Table 1, we draw two conclusions.

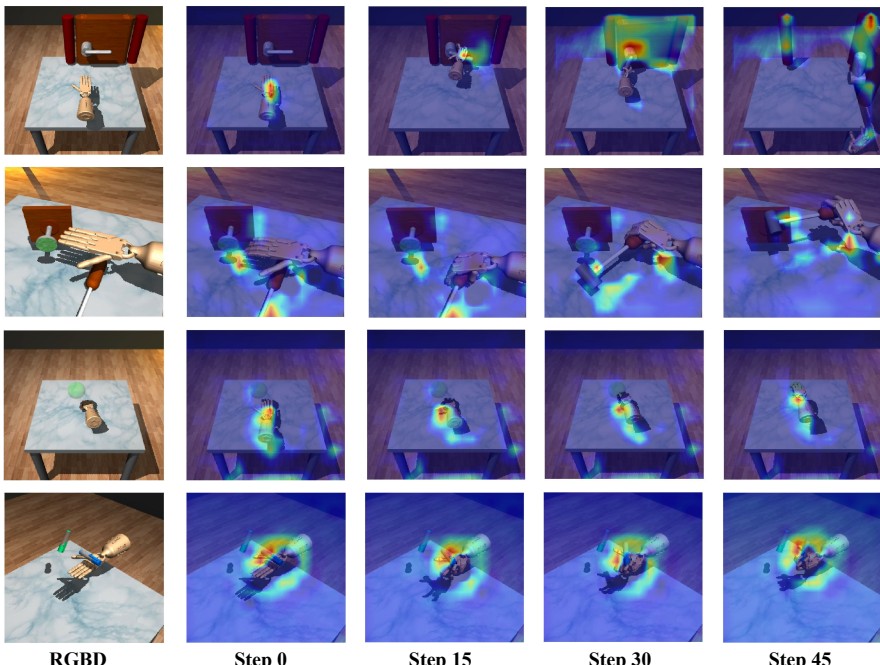

| RGBD | Step 0 | Step 15 | Step 30 | Step 45 |

Figure 3: **Visualization of Object Features Using Class Activation Mapping (CAM) [46].** The sequence from top to bottom illustrates the tasks of door opening, tool use, relocation, and reorientation, while from left to right shows the execution of each task.

*(1) With sufficient data, KOROL with learned feature achieves similar or higher success rate compare to Koopman operator with GT object state and other baselines.* Across all tasks with 200 demonstrations, we notice the difference of KOROL and Koopman operator on easier tasks (Door opening and Tool use) are minimal and the margin magnified on harder tasks (Relocation and Reorientation). KOROL with learned feature achieves 4.4 % and 2.8 % higher success rate on harder tasks respectively. This enhanced performance is attributed to the capability of the learned features to undergo continuous updates during the training of the ResNet model and to adapt dynamically during task execution (see Figure 3). This approach contrasts with using a fixed object state, enhancing KOROL's generality and robustness. In comparison, KOROL with learned features exceeds the model-based NDP across four tasks with an average enhancement of 1.08× and surpasses the learning-based Diffusion Policy by 1.16× when supplied with 200 demonstrations.

*(2) While KOROL's performance diminishes under limited data (10) constraint, it still substantially outperforms other baselines, suggesting it has better sample efficiency.* BC exhibit zero or near-zero performance on most tasks, regardless of whether they use GT object states or learned object features. NDP yield results comparable to KOROL with 200 demonstrations but underperform when reduced to 10, underscoring its dependence on large training datasets. Overall, KOROL exceeds NDP with an average enhancement of 13.77× and surpasses Diffusion Policy by 1.13× when supplied with 10 demonstrations. Our experiments also show that KOROL with learned features exhibits a smaller performance drop (9.5 % average across four tasks) compared to the Koopman operator with GT object state (23.75 %) when reducing demonstrations from 200 to 10. This sample efficiency in KOROL stems from employing the dynamics model—Koopman operator—and learning robust, generalizable object features.

**Object Feature Visualization.** The results in Fig 3 reveal variable focus within the activation maps. Notably, during the door opening task, initial activation predominantly targets the robot's hand, aligning with our training objective to minimize prediction errors in robot state. As the hand approaches the door handle, the activation extends to encompass the hand and the handle. Ultimately, the activation map prominently highlights the handle and the door. In the tool-use task, activation primarily centers on the nail and hammer, whereas in the relocation task, it focuses on the robot's hand. These activation mappings are derived from ResNet18 in training images, specifically

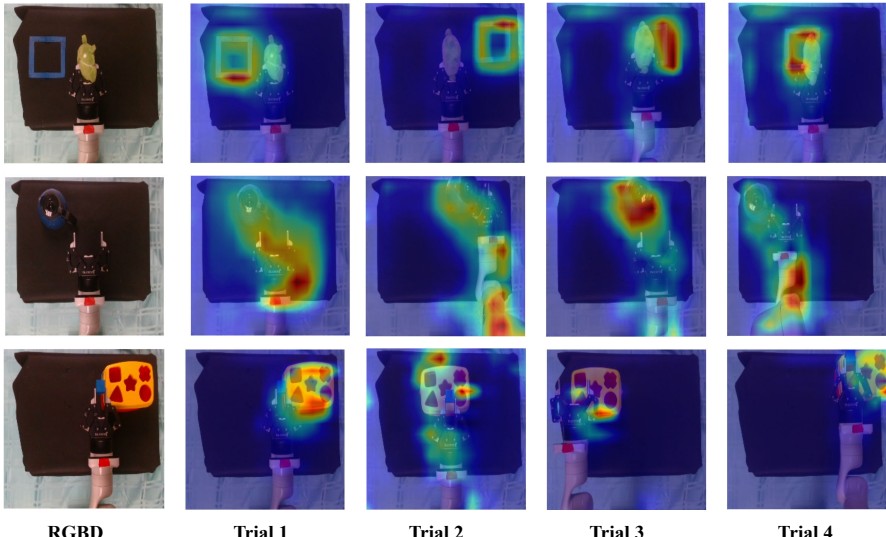

| RGBD | Trial 1 | Trial 2 | Trial 3 | Trial 4 |

Figure 5: **Visualization of Object Features Using CAM in Three Real-World Tasks.** From top to bottom, the sequence showcases training images from various trials of toy relocation, teapot pickup, and cube insertion tasks, demonstrating the feature extractor's generalization to positional variance.

from the output of the last convolutional layer [46]. These activation maps also serve as valuable indicators of the model's training progress. A sufficiently trained KOROL typically exhibits task-relevant feature activation. Conversely, activation focused on irrelevant areas suggests inadequate learning of object features, potentially leading to task failure.

**Effect of Model Update.** We evaluate whether the training of $f_\theta$ depends on the Koopman operator by ablating the update of $\mathbf{K}$ and plotting the training curves in Figure 4. The blue lines show the standard training of KOROL and the orange lines show the ablation. The loss decreases significantly after recalculating $\mathbf{K}$ at epochs 50; otherwise, it remains stagnant. Subsequent updates to $\mathbf{K}$ at epochs 100, 150, and 200 show minimal impact, likely due to the already diminished magnitude of the loss. Additional ablation studies on the performance improvement from using frequency domain images can be found in the Appendix.

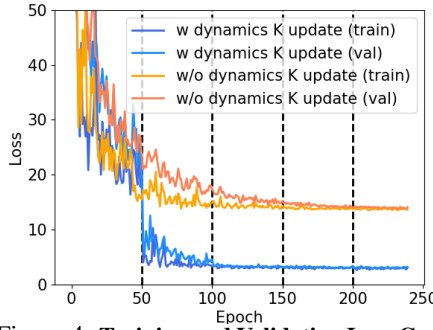

Figure 4: **Training and Validation Loss Curves in *Door* Task**. The dashed line indicates the times of updating $\mathbf{K}$.

### 5.2 Real World Experiment

**Setup.** In our real-world robot experiments, we employed a 7-DoF Kinova robot equipped with a parallel gripper to perform three distinct tasks: (1) *Toy relocation*: Move the green toy on the gripper to a randomized target location (blue bounding box) and release it. (2) *Tea pot pickup*: Grasp the handle of the teapot, which is placed at a randomized position on the table, and lift it up. (3) *Cube insertion*: Move the blue cube on the gripper to a randomized target location (shape sorter box) and drop it into the corresponding shape sorter. We provide 20 and 50 unique demonstrations to each task respectively, and compare KOROL to NDP and Diffusion Policy.

**Numerical Results and Feature Visualization.** KOROL consistently outperforms the baselines, achieving superior average performance (see Table 2). The most frequent failure mode for KOROL involves the gripper moving to a position, typically 1 to 2 cm away from the target, before attempting to grasp the handle or drop the cube. This imprecision results in missing the handle or inaccurately aligning with the shape sorter. In Figure 5, the activation maps of object features delineates the bounding box and the teapot. However, it does not highlight the cube shape sorter, instead emphasizing surrounding areas. This may explain to the lower success rate in the insertion task.

For baseline models, NDP continues to struggle to accurately predict the correct positions in *Pickup* and *Insertion* tasks and doesn't have much performance improvement even with more data. In the *Relocation* task, Diffusion Policy generally succeeds in positioning the gripper correctly but fails to learn the appropriate timing for opening the gripper with only 20 demonstrations. How-

| Task | Relocation | | Pickup | | Insertion | |
|---|---|---|---|---|---|---|
| | 20 | 50 | 20 | 50 | 20 | 50 |
| NDP | 10 | 11 | 0 | 0 | 0 | 0 |
| Diffusion Policy | 0 | 13 | 2 | 7 | 5 | 9 |
| KOROL | **20** | **20** | **17** | **19** | **11** | **14** |

Table 2: **Real-World Manipulation Quantitative Performance.** The number of successful task executions of all models trained with 20 and 50 demonstrations respectively, and evaluated on 20 unique cases per task in real world.

ever, it improves in opening the gripper with additional training data. In *Pickup* and *Insertion*, which require high precision in positional accuracy, Diffusion Policy typically cannot generate sufficiently accurate positions for picking or dropping.

## 5.3 Multi-tasking Experiment

To evaluate the multitasking capabilities of using object features, we combined the training datasets from four tasks into 800 demonstrations and trained a single ResNet model $f_\theta$ alongside a multitasking Koopman operator $\mathbf{K}_{\mathrm{multi}}$. The results in Table 3 reveal that the multitasking Koopman operator sustains robust performance across the *Door opening*, *Tool use*, and *Reorientation* tasks, but exhibits performance declines in the *Relocation* task com-

| Task | ResNet18 | ResNet34 | ResNet50 |
|---|---|---|---|
| Door opening | 99.9% | **100%** | 100% |
| Tool use | **100%** | 99.9% | 100% |
| Relocation | 78.2% | **93.8%** | 81.3% |
| Reorientation | 85.9% | **86.8%** | 85.9% |

Table 3: **Quantitative Performance of KOROL in Multi-Tasking.** The averaged multi-tasking success rates across 5 random seeds of KOROL with ResNet18, ResNet34 or ResNet50 trained with 800 demonstrations and evaluated on 200 unseen cases per task.

pared to KOROL trained with 200 demonstrations per task (see Table 1). Furthermore, the results highlights the need for a feature extractor with substantial capacity to ensure generalizability across tasks. Specifically, the multitasking Koopman operator $\mathbf{K}_{\mathrm{multi}}$ with ResNet34 and ResNet50 improves performance in the *Relocation* task over ResNet18. However, ResNet50 may be too large and thus prone to underfitting, leading to a decline in performance.

## 6 Conclusion

This work introduces and evaluates KOROL, which leverages the Koopman operator rollouts to learn object features for manipulation tasks. KOROL iterative updates the Koopman operator alongside the trained object features to enhance performance. Experiments suggest that KOROL can: (i) improve performance across various simulated manipulation tasks compared to the Koopman operator with GT object state and baseline models, (ii) extend Koopman-based methods to vision-based real-world tasks, and (iii) facilitate multitasking $\mathbf{K}_{\mathrm{multi}}$ with dimensionally-aligned object features.

## 7 Limitations and Future Work

KOROL has several limitations and directions for future research: (1) We currently compute the Koopman operator $\mathbf{K}$ by solving a least-squares problem. Advancements in neural Koopman approaches [7] could allow training the Koopman operator and object features in an end-to-end way. (2) KOROL underperforms in fine-grain manipulation tasks, such as cube insertion. Future work could focus on refining object feature accuracy and enhancing control precision using more advanced feature extractors, such as vision transformers [47]. (3) The CAM visualization technique for object features is restricted to spatial domain RGBD images and is not applicable to frequency domain images. Currently, we verify object feature accuracy through CAM visualization and test model performance using RGB-D images before incorporating frequency domain images to enhance performance, albeit without visualization. Exploring visualization techniques for frequency domain images represents a promising avenue for future research.

**Acknowledgments**

The authors would like to thank the anonymous reviewers for their insightful feedback, which has helped improve the quality of this paper. We are grateful to Abulikemu Abuduweili, Aviral Agrawal, and Yunhai Han for their valuable discussions and contributions throughout the project. Special thanks go to our advisors, Harish Ravichandar, Changliu Liu, and Jeffrey Ichnowski, for their continuous guidance and mentorship. We also acknowledge the computing resources and robotic infrastructure provided by Changliu Liu's Intelligent Control Lab at Carnegie Mellon University.

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

# Appendix

## A    ADROIT Hand Experimental Details

### A.1    Task State Space Design

**Door opening.**    Given a randomized door position, undo the latch and drag the door open. In this task, $x_r(t) \in \mathcal{X}_r \subset \mathbb{R}^{28}$ (24-DoF hand + 3-DoF wrist rotation + 1-Dof wrist motion) as the floating wrist base can only move along the direction that is perpendicular to the door plane but rotate freely. Regarding the object states, $x_o(t) = [p_t^{\text{handle}}, v_t, p^{\text{door}}] \in \mathcal{X}_o \subset \mathbb{R}^7$, containing the door position $p^{\text{door}}$, handle position $p^{\text{handle}}$ and the angular velocity of the door opening angle $v_t$. In each test case, we randomly sampled door positions $p^{\text{door}}$ $(xyz)$ from uniform distributions: $x \sim \mathcal{U}(-0.3, 0)$, $y \sim \mathcal{U}(0.2, 0.35)$, and $z \sim \mathcal{U}(0.252, 0.402)$.

**Tool use.**    Pick up the hammer to drive the nail into the board placed at a randomized height. In this task, $x_r(t) \in \mathcal{X}_r \subset \mathbb{R}^{26}$ (24-DoF hand + 2-DoF wrist rotation) as the floating wrist base can only rotate along the $x$ and $y$ axis. $x_o(t) = [p_t^{\text{tool}}, o_t^{\text{tool}}, p^{\text{nail}}]$ containing the nail goal position $p^{\text{nail}}$, hammer positions $p_t^{\text{tool}}$ and orientations $o_t^{\text{tool}}$. In each test case, we randomly sampled nail height $(z)$ in $p^{\text{nail}}$ from a uniform distribution: $z \sim \mathcal{U}(0.1, 0.25)$.

**Object relocation.**    Move the blue ball to a randomized target location (green sphere). In this task, $x_r(t) \in \mathcal{X}^r \subset \mathbb{R}^{30}$ (24-DoF hand + 6-DoF floating wrist base) as the ADROIT hand is fully actuated. $x_o(t) = [p_t^{\text{ball}}, o_t^{\text{ball}}]$ containing the target positions $p^{\text{target}}$ and current positions $p_t^{\text{ball}}$. In each test case, we randomly sampled target positions $p^{\text{target}}$ $(xyz)$ from uniform distributions: $x \sim \mathcal{U}(-0.25, 0.25)$, $y \sim \mathcal{U}(-0.25, 0.25)$, and $z \sim \mathcal{U}(0.15, 0.35)$.

**In-hand reorientation.**    Reorient the blue pen to a randomized goal orientation (green pen). In this task, $x_r(t) \in \mathcal{X}_r \subset \mathbb{R}^{24}$ (24-DoF hand) as floating wrist base is fixed. $x_o(t) = [p_t^{\text{pen}}, o_t^{\text{pen}}]$ containing the goal orientations $o^{\text{goal}}$ and current pen orientations $o_t^{\text{pen}}$, which are both unit direction vectors. In each test case, we randomly sampled the pitch $(\alpha)$ and yaw $(\beta)$ angles of the goal orientation $o^{\text{goal}}$ from uniform distributions: $\alpha \sim \mathcal{U}(-1, 1)$ and $\beta \sim \mathcal{U}(-1, 1)$.

The task success criteria is the same as defined in [6].

### A.2    Policy Design and Training

**Koopman Operator**    The lifting functions of Koopman Operator are taken from [6]. The representation of the system is given as: $x_r = [x_r^1, x_r^2, \cdots, x_r^n]$ and $x_o = [x_o^1, x_o^2, \cdots, x_o^m]$ and superscript is used to index states. In experiments, the vector-valued lifting functions $\psi_r$ and $\psi_o$ in (3) were defined as polynomial basis functions:

$$\begin{aligned} \psi_r &= \{x_r^i x_r^j\} \cup \{(x_r^i)^2\} \cup \{(x_r^i)^3\} \text{ for } i, j = 1, \cdots, n \\ \psi_o &= \{x_o^i x_o^j\} \cup \{(x_o^i)^2\} \cup \{(x_o^i)^2 (x_o^j)\} \text{ for } i, j = 1, \cdots, m \end{aligned} \tag{7}$$

Note that $x_r^i x_r^j / x_r^j x_r^i$ and $x_o^i x_o^j / x_o^j x_o^i$ each appear only once in the lifting functions. $t$ is ignored here as the lifting functions are the same across the time horizon. Thus, the dimension of the Koopman Operator $\mathbf{K} \in \mathbb{R}^{p \times p}$, where $p = 3n + 2m + m^2 + \frac{n(n-1)}{2} + \frac{m(m-1)}{2}$.

**KOROL Training**    In *Door opening* and *Tool use* tasks, the feature extractor is trained solely using RGBD images. While in *Relocation* and *Reorientation* tasks, the feature extractor is additionally provided with the desired goal locations $p^{\text{target}}$ and goal orientations $o^{\text{goal}}$. The full list of training hyperparameters can be found in Table 4.

**Effect of Model Update Frequency**    We analyze the impact of hyperparameter M, the frequency of updating $\mathbf{K}$, on KOROL's performance in Table 5. Without model updating, the success rate is zero because the Koopman operator $\mathbf{K}$ becomes outdated for the trained object features. Conversely, excessively frequent updates to $\mathbf{K}$ destabilize object feature training and degrade performance, similar to the target policy update delay in Twin Delayed DDPG (TD3) [48]. Therefore, the update

| Hyperparameter | Value |
|---|---|
| Feature Extractor | ResNet18 |
| Input RGBD Image Dimension | $256 \times 256 \times 4$ |
| Input Desired Poisition and Orientation Encoder | HarmonicEmbedding |
| Input Desired Poisition and Orientation Dimension | 3 |
| Output Desired Poisition and Orientation Embedding Dimension | 15 |
| Output Object Feature Dimension | 8 |
| Batch Size | 8 |
| Prediction Horizon | 40 |
| Learning rate | $1 * 10^{-4}$ |
| Adam betas | $(0.9, 0.999)$ |
| Learning rate decay | Linear decay (see code for details) |
| Max Training Epoch | 300 |
| Max Execution Step Num | 100 |

Table 4: **Hyperparameters of KOROL Training for ADROIT Hand Experiments.**

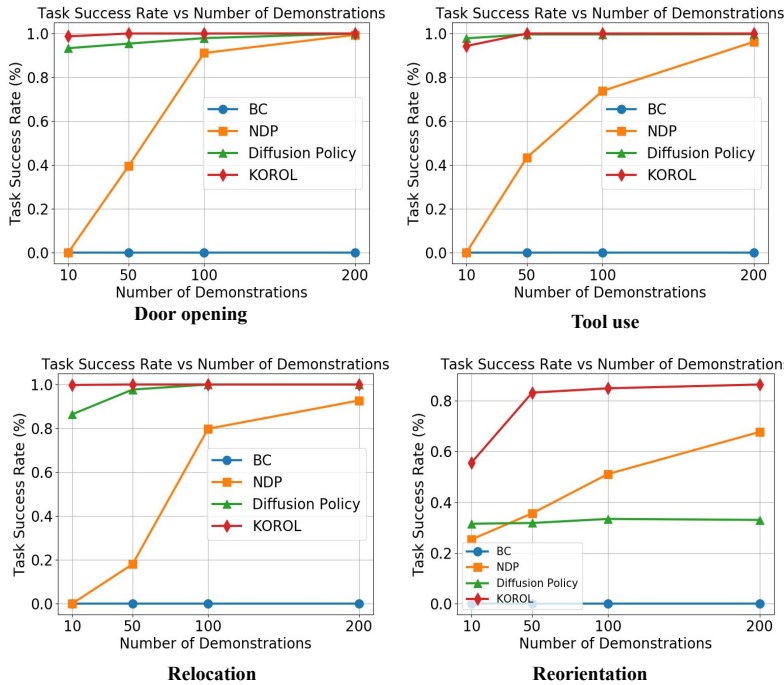

Figure 6: **The effects of number of demonstrations on success rate for all models on each task.**

frequency of the Koopman operator $\mathbf{K}$ should be much lower than that of the object feature training, usually by at least an order of magnitude.

| M (Number of epochs for $\mathbf{K}$ updating) | 1 | 10 | 20 | 50 | 100 | $\infty$ |
|---|---|---|---|---|---|---|
| Num of $\mathbf{K}$ Update | 100 | 10 | 5 | 2 | 1 | 0 |
| Success rate | 80.4% | 87.2% | 96.2% | 99.9% | 99.9% | 0% |

Table 5: **KOROL Performance in Door Opening Task After 100 Training Epochs with Varied K Update Frequencies.**

**Effect of Number of Demonstrations**    To investigate scalability and sample efficiency, we trained all models on varying numbers of demonstrations (10, 50, 100, 200) and plot the success rate on 200 unseen cases per task at Figure 6. KOROL consistently achieves the highest task success rate in most scenarios, except for the Tool use task with 10 demonstrations.

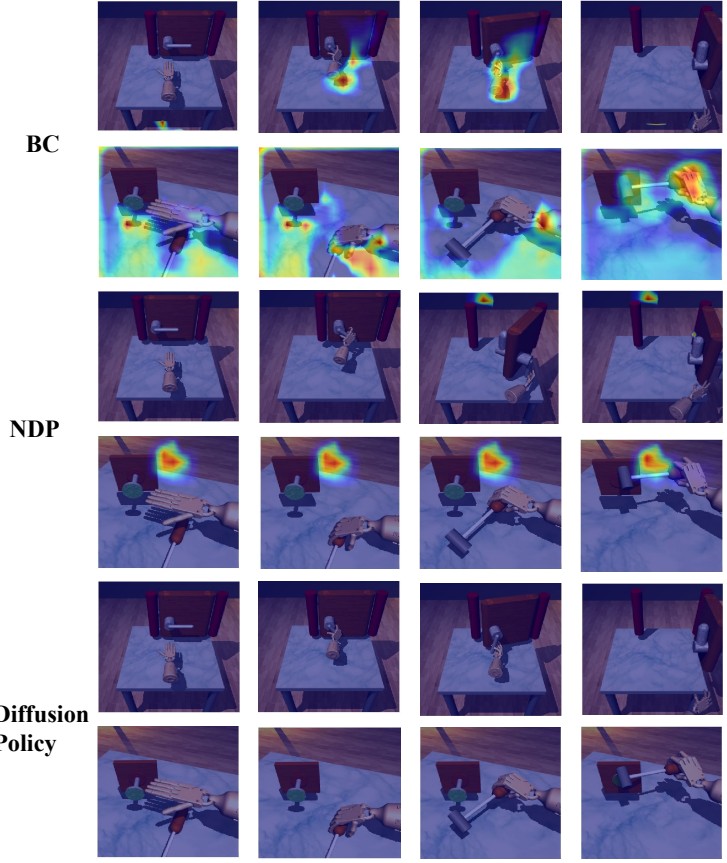

Figure 7: **Visualization of Object Features from Baselines Using CAM in Door opening and Tool use Tasks.** From top to bottom, the sequence displays visualization images from BC, NDP, and Diffusion Policy.

## A.3 Baselines

We ran BC and NDP based on the implementation in [6]

https://github.com/GT-STAR-Lab/KODex.

For Diffusion Policy, we used the author's original implementation [16]

https://github.com/real-stanford/diffusion_policy.

We present the object features visualizations of these baselines in Door Opening and Tool Use tasks at Figure 7. These tasks involve several objects, including the robot hand, door, handle, hammer, and nails, which require special attention and are easier to interpret compared to Relocation and Reorientation. The results show that the object features learned from BC are more readable than those from NDP and Diffusion Policy, but not as good as KOROL's. This difference may be because KOROL and BC maintain relatively simple model structures that preserve the visual interpretability of object features, while the Diffusion Policy obscures this information due to its complex denoising process. However, BC's model power is too low (0% success rate in tasks), and thus the features it learns do not fully capture the essential information of the scenes for manipulation. In contrast, KOROL's features effectively highlight the door handle, nail, and hammer.

### A.4 Inverse Dynamic Controller

We employ a pre-trained inverse dynamics controller $C$, specific to each task, as detailed in [6]. Each controller $C$ is trained to output actions corresponding to the dimensionality of the robot state defined for its specific task.

## B Real-World Experimental Details

### B.1 Robot State Space and Task Definition

In the physical robot experiment, we employ a Kinova robotic arm. The configuration space of the robot $x_r(t) \in \mathcal{X}_r \subset \mathbb{R}^7$ includes three degrees of freedom (DOF) for the end-effector's position, three DOF for its orientation (ranging from 0 to 360 degrees), and one DOF for the gripper's position (ranging from 0 to 1). The task definition and success criteria are discussed in Section 5.2.

### B.2 Experiment Details

The Koopman Operator design, KOROL and baselines training are the same as in our simulation. The only difference is that we no longer need to use an inverse dynamic controller to compute torque for each joint. Instead, we publish the predicted end-effector position and gripper position through Kinova API to control robot.

| Model | Door opening | | | Tool use | | | Relocation | | | Reorientation | | |
|---|---|---|---|---|---|---|---|---|---|---|---|---|
| | 10 | 50 | 200 | 10 | 50 | 200 | 10 | 50 | 200 | 10 | 50 | 200 |
| BC w/o | 0% | 0% | 0% | 0% | 0% | 0% | 0% | 0% | 0% | 0% | 0% | 0% |
| NDP w/o | 0% | 39.5% | 99.3% | 0% | 43.4% | 96.2% | 0% | 18.0% | 92.7% | 25.3% | 35.6% | 67.7% |
| Diffusion Policy w/o | 93.2% | 95.3% | 99.9% | 97.8% | 99.6% | 99.7% | 86.4% | 97.7% | 100% | 31.5% | 31.8% | 33.0% |
| KOROL w/o | 93.2% | 95.7% | 99.9% | 84.5% | 100% | 100% | 45.5% | 98.4% | 100% | 17.4% | 82.7% | **87.0%** |
| BC w | 0% | 0% | 0% | 0% | 0% | 0% | 0% | 0% | 0% | 0% | 0% | 0% |
| NDP w | 0% | 87.6% | 95.2% | 0% | 65.4% | 89.2% | 0% | 27.8% | 100% | 14.1% | 26.3% | 27.9% |
| Diffusion Policy w | 74.2% | 72.1% | 66.1% | 51.7% | 100% | 99.9% | 90.9% | 96.9% | 100% | 30.9% | 30.5% | 38.6% |
| KOROL w | **98.6%** | **99.9%** | **99.9%** | **94.3%** | **100%** | **100%** | **99.8%** | **100%** | **100%** | **55.6%** | **83.2%** | 86.4% |

Table 6: **KOROL and Baselines Performance in ADROIT Hand with and w/o Frequency Domain Image, trained with 10, 50 and 200 demonstrations per task.**

| Task | Relocation | Pickup | Insertion |
|---|---|---|---|
| KOROL w/o | 19/20 | 17/20 | 6/20 |
| KOROL w | 20/20 | 19/20 | 11/20 |

Table 7: **KOROL Performance in Real-World Manipulation with and w/o Frequency Domain Images.**

| Task | KOROL w/o transformation | | | KOROL | | |
|---|---|---|---|---|---|---|
| | ResNet18 | ResNet34 | ResNet50 | ResNet18 | ResNet34 | ResNet50 |
| Door opening | 99.9% | 96.0% | 0% | 99.9% | 100% | 100% |
| Tool use | 75.3% | 48.9% | 0% | 100% | 99.9% | 100% |
| Relocation | 49.1% | 91.6% | 0% | 78.2% | 93.8% | 81.3% |
| Reorientation | 86.6% | 85.3% | 23.8% | 85.9% | 86.8% | 85.9% |

Table 8: **KOROL Performance in Multi-tasking Tasks with and w/o Frequency Domain Images.**

## C Multi-tasking Experimental Details

As discussed in Section A, the robot state space in the Mujoco environment varies slightly across different tasks. To standardize this, we augment the state space to $\mathbb{R}^{30}$, which includes a 24-DoF

hand and a 6-DoF floating wrist base, by padding zeros to the missing robot states. For instance, in *Door opening* task, we pad zeros to the $Tx$ and $Ty$ motion directions.

For multi-tasking controllers, it is necessary to remove the padding from the robot state and select the appropriate elements to compute the action accordingly. When evaluating the unified Koopman operator $\mathbf{K}$ and the feature extractor $f_\theta$, we continue to use a specific controller $C$ for each task due to time constraints. However, we believe it is entirely feasible to train a single, unified controller $C$ for all tasks with dimensionally-aligned demonstrations.

## D   Ablation of Using Image Transformation

Because of the enhanced performance observed in prior works  [42, 43] using frequency domain images, this section evaluates the impact of employing transformed images in the frequency domain across various settings: simulation, real-world manipulation, and multi-tasking. The model denoted as KOROL utilizes both spatial and frequency-domain images as inputs, whereas KOROL w/o transformation uses only spatial images. The results in Table 6, Table 7 and Table 8 demonstrate significant of KOROL improvements achieved by incorporating transformed images in all tasks, corroborating the findings in [42, 43]. In detail, learning in the frequency domain helps a neural network to learn richer features leading to a higher distinguishing power between inputs with high similarity. For example, in the image observations of Door opening task, high-frequency components in the transformed input capture the slight change of position of the door and the robot hand. At the same time, the low-frequency components capture the general scene elements that remain largely unchanged, like the background. In Table 6, we observed similar performance improvements in NDP with 50 demonstrations, but a drop in Diffusion Policy across most cases (10, 50, 200 demonstrations). We noted that both NDP and KOROL maintain a relatively simple structure by using object features as input to construct a dynamical system, which is used for predicting robot trajectories. However, in Diffusion Policy, the features are used as inputs for further neural network computations (e.g., UNet denoising) that generate downstream features for their respective predicting modules. While DCT-based features help in creating richer features, subsequent neural feature extraction requires further experimentation to work effectively in tandem with DCT features.

