# OpenReview forum: "KOROL: Learning Visualizable Object Feature with Koopman Operator Rollout for Manipulation"
_robot-learning.org/CoRL/2024/Conference — CoRL 2024_

### Official Review · Reviewer_wypC · 2024-07-18
**KOROL**

**Originality:** 4
**Technical Quality:** 4
**Clarity Of Presentation:** 5
**Potential Impact:** 3
**Recommendation:** 4
**Confidence:** 4

**Review:**

Strengths:
- The paper proposes an exciting method for scaling Koopman operator learning  to high-dimensional observations for dexterous manipulation tasks, a critical challenge in enabling model-based control in realistic real-world conditions.
- The move to visual observations allows for multi-task learning in a unified observation space.
- The paper is clearly written and provides a concise, self-contained introduction to Koopman theory.
- The method is technically sound and design decisions are justified with ablations.
- The evaluation is comprehensive and demonstrates improvement over state-of-the-art imitation learning algorithms in sim and real tasks.

Weaknesses:
- The scientific contribution is relatively minor over the previous work in KODEX; that said, I think there is enough technical detail and experimental evaluation to support this as a standalone paper.
- The imitation objective is defined over robot state only. Is this always sufficient? It would be interesting to see more complex observation spaces like force/torque and tactile data which may be necessary for more dexterous tasks. Would the approach scale to these observations? Should they also be encoded into a learned space rather than regressed directly? Some additional simulation experiments in this direction could be interesting.
- The number of demonstration trajectories in the real experiments seems arbitrary. Using a more standard quantity (i.e., 50-100) and performing the same evaluation scheme as in sim (i.e., to quantify the effect of more data) would strengthen the argument.

**Quality Of The Limitations Section:**

3

**Questions For Rebuttal:**

Please see weaknesses above; in particular, some discussion and/or experiments about additional sensing modalities, and perhaps additional real-world comparison with differing amounts of data could strengthen the argument.

**Robotics Focus:**

4

**Summary Of Paper:**

The paper proposes a method for learning Koopman operators from high-dimensional observations in the context of dexterous manipulation tasks, enabling model-based control without the assumption of low-dim states. The central contributions are an algorithm and system architecture for jointly learning object feature encoders jointly with the Koopman operator from demonstrations. The model rollouts can then be tracked with a learned inverse dynamics controller from previous work. The experimental evaluation demonstrates the proposed approach outperforms state-of-the-art imitation learning algorithms on simulated Adroit tasks and real pick-and-place tasks.

**Summary Of Recommendation:**

Overall, the paper studies an important problem, offers an exciting solution, and provides thorough experimentation and analysis.

---

### Official Review · Reviewer_aTms · 2024-07-19
**Well-executed work with certain limitation**

**Originality:** 4
**Technical Quality:** 3
**Clarity Of Presentation:** 4
**Potential Impact:** 3
**Recommendation:** 3
**Confidence:** 4

**Review:**

Strength:

1. I think the overall idea of learning visual features and applying Koopman operation is intuitive and well executed. The experiment results show significant improvement in BC success rate compared to strong baselines such as NDP and Diffusion Policy. (To be honest and no discredit to the authors, as someone who has not followed Koopman literature in robotics, I am surprised that this idea has not been explored before.)

2. Multi-task positive transfer shown in 5.3 is well appreciated — I think this is a missing piece of many recent BC literature. I also think more multi-task experiments with baselines would be great. I wonder if Koopman Operator encourages more positive transfer than baselines.

Limitation:

1. As the authors mention, KOROL underperforms in fine-grained manipulation tasks. To me this is an inherent limitation of using linear Koopman Operator, which limits the expressivity of the policy. My intuition is that on more challenging real-world tasks KOROL might not be scalable.

2. I am not sure about the object feature visualizations — since there is no comparison to features learned from the baselines, I don’t feel Fig. 4 and 5 help illustrate the benefit of KOROL much.

**Quality Of The Limitations Section:**

2

**Questions For Rebuttal:**

DCT is critical to the high success rate of KOROL as Table 5 in the appendix shows. Could you provide any explanation if you have any? I wonder if DCT brings some benefit that is specific to KOROL. Also for a fair comparison, DCT should also be applied to the baselines.

Could you comment on how you can better handle more fine-grained manipulation tasks with some extension of KOROL?

Multi-modality has been known to help BC generalize and be robust to perturbations — Does KOROL also handle multi-modality to some extent?

About Figure 4: is it reasonable to consider validation loss at all with learning Koopman operator? It is possible the model is overfitting with the training loss drop. Or the magnitude is too large for any overfitting?

**Robotics Focus:**

4

**Summary Of Paper:**

This paper introduces KOROL, a behavior cloning method that learns visual features that observe the Koopman Operator in the feature space. KOROL learns a visual extractor and also the Koopman matrix at the same time. KOROL shows relatively superior performance than model-based and model-free baselines in ADROIT and three real-world tasks.

**Summary Of Recommendation:**

This is a well-executed paper with a novel idea, despite certain limitations. I recommend acceptance of the paper, and I hope the authors further address the limitations in the rebuttal.

---

### Official Review · Reviewer_ekAf · 2024-07-27
**Simple yet interesting method but experiments need to be improved**

**Originality:** 3
**Technical Quality:** 3
**Clarity Of Presentation:** 4
**Potential Impact:** 3
**Recommendation:** 3
**Confidence:** 4

**Review:**

## Comments
+ The method is simple. The core contribution is to use a learned model in visual encoding. It’s nice having a learned visual feature extractor to replace the low-level object states. Together with the proposed training scheme, KOROL makes the Koopman operator more applicable in real-world settings.
+ Section 3 is well-written and provides a nice walk-through for readers not familiar with the Koopman operator. The equations are clear and easy to follow.
- I do think the experiments need to be improved, or the claims should be revised. Firstly, the experiments focus a lot on ‘sample efficiency’ of the Koopman operator from my view. It was compared to baselines using different numbers of training data. However, sample efficiency is not emphasized in previous sections, but only shows up as an analysis of the experimental result. It would be better to include discussions in previous sections on this point. Secondly, if the core contribution of KOROL is removing the need for GT low-level object state,  an important baseline will be trying to use perception modules to extract object state from images, and show that KOROL does better than that.
- I don’t understand why learned visual encoding will be better than GT object state. The GT state should always contain more information than a learned visual feature if all relevant low-level states are included in the GT state. Does the designed `GT object state' include all relevant information? e.g. according to the appendix, in tool use, only the goal position of the nail is in the object state, but not the current state of the nail.


## Misc
- I didn’t realize that K is computed by solving the least-squares problem, and I was confused about why not updating K more frequently. It could be helpful to clarify that earlier.
- For clarity, it’s better to swap 5.2 and 5.3 in my opinion, and somehow shows the original performance in Table 3. Right now it’s difficult to compare the success rate of 800 demonstrations to 200 demonstrations based on Table 3.
- Varying M (epoch num to update K) could be an interesting ablation.

**Quality Of The Limitations Section:**

3

**Questions For Rebuttal:**

- It wasn’t mentioned how many testing episodes are used. And I would love to see some numbers about variance since some numbers are really close.
- Does the performance plateau at 200 demonstrations? It would be nice to have a plot so that the sample efficiency can be compared in a more straightforward way.
- Why is Koopman having a big performance drop (compared to diffusion policy) on the reorientation task?

**Robotics Focus:**

4

**Summary Of Paper:**

This paper proposes to use Koopman operator with learned visual features directly from images, named KOROL. Koopman operator has been previously used with low-level object state, which makes it difficult to be applied in the real world. The proposed method uses a learned visual feature extracted from RGBD images. KORAL is evaluated on both sim and real data.

**Summary Of Recommendation:**

Interesting application but the experiments need to be strenghthened. I'm inclined to rejection for the current version.

---

### Author Rebuttal · Authors · 2024-08-08

We thank all Reviewrs' careful review and suggestions. We have added new ablation experiments and explanations to address each of your concerns and have uploaded the latest version of the paper accordingly, with all changes highlighted in red.

---

### Decision · Program_Chairs · 2024-09-04

**Decision:**

Accept

**Comment:**

This paper uses a koopman operator to learn interpretable visual feature for manipulation. In general, reviewers leaned towards positive on this work, though had some outstanding queries to be addressed in the rebuttal.

Strengths:
* Simple model, well explained
* Good multi-task transfer experimentation

Weaknesses:
* Some mismatch between claims and experiments, consider revising to align with experiments actually carried out
* Relatively minor improvements over prior work (eg. KODex)
* Unclear where limitations are, given experimental settings
* Some questions remain about experimental settings

Post rebuttal:
Authors provided a strong rebuttal, including additional experimentation results and ablations. Reviewers are in agreement that this paper should be accepted.